

# Improving Ozone Simulations in Asia via Multisource Data Assimilation: Results from an Observing System Simulation Experiment with GEMS Geostationary Satellite Observations

Lei Shu[1,2], Lei Zhu[2,3], Juseon Bak[4], Peter Zoogman[5], Han Han[1], Song Liu[2], Xicheng Li[2], Shuai Sun[2], Juan Li[2], Yuyang Chen[2], Dongchuan Pu[2], Xiaoxing Zuo[2], Weitao Fu[2], Xin Yang[2,3], and Tzung-May Fu[2,3]

[1]School of Geographical Sciences, Fujian Normal University, Fuzhou, Fujian 350007, China
[2]School of Environmental Science and Engineering, Southern University of Science and Technology, Shenzhen, Guangdong 518055, China
[3]Guangdong Provincial Observation and Research Station for Coastal Atmosphere and Climate of the Greater Bay Area, Shenzhen, Guangdong 518055, China
[4]Institute of Environmental Studies, Pusan National University, Busan 46241, South Korea
[5]Harvard-Smithsonian Center for Astrophysics, Cambridge, Massachusetts 02138, United States

*Correspondence to*: Lei Zhu (zhul3@sustech.edu.cn)

**Abstract.** The applications of geostationary (GEO) satellite measurements at an unprecedented spatial and temporal resolution from the Geostationary Environment Monitoring Spectrometer (GEMS) for monitoring and forecasting the alarming ozone pollution in Asia through data assimilation remain at the early stage. Here we investigate the benefit of multiple ozone observations from GEMS geostationary satellite, low Earth orbit (LEO) satellite, and surface networks on summertime ozone simulations through individual or joint data assimilation, built on our previous Observing System Simulation Experiment (OSSE) framework (Shu *et al.*, 2022). We find that data assimilation better represents the exceedance, spatial patterns, and diurnal variations of surface ozone, with a regional mean negative bias reduction from 2.1 to 0.2–1.2 ppbv in ozone simulations as well as precision improvements of a root-mean-square error (RMSE) of by 5–69 % in most Asian countries. Furthermore, the joint assimilation of GEMS and surface observations performs the best. GEMS also brings direct added value for better reproducing ozone vertical distributions, especially in the middle to upper troposphere at low latitudes, but may mask the added value of LEO measurements, which are crucial to constrain surface and upper tropospheric ozone simulations when observations from other platforms are inadequate. Our study provides a valuable reference for ozone data assimilation as multisource observations become gradually available in the era of GEO satellites.

## 1 Introduction

Ozone is a secondary air pollutant formed in a complex chemical interaction between nitrogen oxides ($NO_x$) and volatile organic compounds (VOCs) in the presence of sunlight. Long-term exposure to ozone increases the risk of premature mortality from respiratory causes, for example, reportedly contributing to an estimated 365,000 deaths from the chronic





obstructive pulmonary disease worldwide in 2019 (Health Effects Institute, 2020). In particular, the majority (~ 80 %) of estimated ozone-attributable respiratory deaths are in Asia, predominantly in India and China (Malley *et al.*, 2017), where operational monitoring sites are spatially scattered and sparse. As the first geostationary air quality monitoring satellite

instrument, the Geostationary Environment Monitoring Spectrometer (GEMS) onboard the Geostationary Korea Multi-Purpose satellite (Geo-KOMPSAT)-2B continuously measures tropospheric ozone in the daytime over East Asia (Kim *et al.*, 2020), providing unprecedented opportunities to fill the observational gaps over this region. However, the extent to which the observing system during the operation of GEMS, principally consisting of measurements from GEMS, low Earth orbit (LEO) satellites, and ground-based stations, would improve ozone simulations and forecasting remains unclear. We quantify

here the added value of individually- or jointly-assimilating multiple ozone observations for improving ozone simulations in Asia, built on our previous Observing System Simulation Experiment (OSSE) framework (Shu *et al.*, 2022).

Ozone pollution in Asia has been worsening in the last decade, resulting from rapid urbanization and industrialization (Gaudel *et al.*, 2018; Chen *et al.*, 2021). In East Asia, lower tropospheric ozone column measurements reveal a decadal

increase of 0.21±0.05 Dobson Unit (DU) during 2005–2018 in eastern China, South Korea, and Japan (Lee *et al.*, 2021). Near the surface, the observed daily maximum 8-h average (MDA8) ozone concentrations have been steadily increasing by ~ 4–8 % yr$^{-1}$ during 2013–2020 according to the Chinese surface network (Han *et al.*, 2020; Lu *et al.*, 2020), along with intensified persistent ozone pollution episodes lasting for 5 days or longer (Gong *et al.*, 2020; Shu *et al.*, 2020). Ground-based observations in South Korea likewise indicate a nationwide increase of 0.3 to 1.7 ppbv yr$^{-1}$ in surface MDA8 ozone

during 2001–2018 (Yeo and Kim, 2021). In Southeast Asia, elevated MDA8 ozone levels exceeding the recommended exposure of 50 ppbv (Assareh *et al.*, 2016; Marvin *et al.*, 2021) are reported from biomass burning (Reddington *et al.* 2021; Sonkaew and Macatangay, 2015; Yadav *et al.*, 2017), often recurringly during the dry season (Ziemke *et al.*, 2009), causing deteriorated ozone pollution in downstream areas (Deng *et al.*, 2008; Lin *et al.*, 2013). In South Asia, countries have observed the steepest increase in ozone exposure globally. For example, India has witnessed an increase of about 17 % in

population-weighted average MDA8 ozone concentrations, ramping from 56.5 ppbv in 2010 to 66.2 ppbv in 2019 (Health Effects Institute, 2020).

The alarming ozone pollution in Asia calls for better understanding and forecasting from air quality models. In particular, data assimilation, incorporated into chemical transport models (CTMs), maximizes the value of observations, thus reducing

biases in ozone numerical simulations (Bocquet *et al.*, 2015; Wu *et al.*, 2008; Huang *et al.*, 2015). Data assimilation utilizes a wide range of observations, generally involving measurements from ground-based stations (Tang *et al.*, 2011; Peng *et al.*, 2018; Ma *et al.*, 2019), LEO satellites (Huang *et al.*, 2015; Miyazaki *et al.*, 2012, 2020b; Inness *et al.*, 2015, 2019b), and geostationary satellites (Claeyman *et al.*, 2011; Zoogman *et al.*, 2011, 2014; Quesada-Ruiz *et al.* 2020; Shu *et al.*, 2022) in the current and anticipated observing system. As such, data assimilation applications have propelled to the forefront of the

development of tropospheric and surface ozone reanalysis, *e.g.*, Monitoring Atmospheric Composition and Climate (MACC)

reanalysis (Inness *et al.*, 2013), Copernicus Atmosphere Monitoring Service (CAMS) interim reanalysis (Flemming *et al.*, 2017), CAMS reanalysis (Inness *et al.*, 2019a), Tropospheric ozone reanalysis (TCR-1/TCR-2) (Miyazaki *et al.*, 2015, 2020a), and Chinese air quality reanalysis (CAQRA) (Kong *et al.*, 2021).

Widely recognized, the value of the measurements from LEO satellites, *e.g.*, Ozone Monitoring Instrument (OMI) (Levelt *et al.*, 2018), Global Ozone Monitoring Experiment-2 (GOME-2) (Munro *et al.*, 2016), and TROPOspheric Monitoring Instrument (TROPOMI) (Veefkind *et al.*, 2012), for improving tropospheric chemistry modeling is immense due to the high spatial coverage (Inness *et al.*, 2019a, 2019b). For example, Miyazaki *et al.* (2020b) demonstrated that the assimilation of multiconstituent LEO satellite data reduces annual mean ozone bias by 39–97 % in the middle troposphere. Sekiya *et al.*
(2022) manifested that the assimilation of TROPOMI tropospheric nitrogen dioxide ($NO_2$) column retrievals leads to improved agreements (by 7–40 %) with independent validation observations compared to control simulation, which is more obvious than those by assimilating OMI data (by 1–22 %) for many cases, owing to the higher spatial resolution and smaller observation errors of TROPOMI measurements.

Observing air quality from space is entering a new era of geostationary satellites, marked by the launch of GEMS on February 2020. Compared with LEO satellites, geostationary satellites can monitor the diurnal variations in tropospheric ozone and its precursors during the daytime. A virtual geostationary constellation, consisting of GEMS over East Asia (Kim *et al.*, 2020), Tropospheric emissions: Monitoring of pollution (TEMPO; 2023) over North America (Zoogman *et al.*, 2017), and Sentinel-4 (2024) over Europe (Ingmann *et al.*, 2012), focuses on the most polluted and industrialized regions in the
Northern Hemisphere. Their value for atmospheric chemistry modeling has been investigated preliminarily through the OSSE approach (Zoogman *et al.*, 2014; Quesada-Ruiz *et al.* 2020; Shu *et al.*, 2022).

GEMS is an ultraviolet (UV) and visible (Vis) imaging spectrometer that facilitates tropospheric ozone monitoring at an unprecedented spatial ($7 \times 8$ km$^2$ at Seoul) and temporal (8 times per day at least) resolution in a spectral range of 300–340
nm over East Asia, yielding massive observations for data assimilation applications (Kim *et al.*, 2020; Bak *et al.*, 2013, 2019). An OSSE presented in our previous work (Shu *et al.*, 2022) has revealed that GEMS could provide useful information to constrain surface and middle-to-upper tropospheric ozone simulations. However, there remains a need for more investigations involving multisource data assimilation. Here we examine to what extent the assimilation of multiple ozone observations (GEMS, LEO satellite, and surface observations), individually or in combination, would improve ozone
simulations in Asia through a robust OSSE as described below.



## 2 Observing System Simulation Experiment (OSSE)

OSSE is a critical tool for objectively assessing the added value of proposed satellite observations to an existing observing system and investigating the impact of instrument designs (Timmermans *et al.*, 2015; Brasseur and Jacob, 2017). In general, the air quality OSSE framework involves: (1) application of a CTM to generate time-varying 3-D fields of atmospheric compositions (taken as the "true" atmosphere), called *nature run*; (2) virtual sampling of such "true" atmosphere following the observing schedules and error characteristics of proposed instruments, that is, to generate synthetic observations; (3) application of a second, preferably independent, CTM to obtain *a priori* and *a posteriori* estimates of atmospheric compositions (without and with the assimilation of synthetic observations), called *control run* and *assimilation run*, respectively; and (4) quantification of the benefit of proposed instruments by examining the correction of mismatch between the "true" state and the *a priori* after data assimilation (Zoogman *et al.*, 2014). In Fig. S1, we sketch the abovementioned steps.

We perform our OSSE for June 2020 to represent summertime ozone in Asia. The observing system includes the GEMS geostationary satellite, an LEO satellite instrument (*i.e.*, OMI), and the surface monitoring network. This study only simulates synthetic OMI ozone profile retrievals to represent the LEO satellite measurements. In the *nature run*, we use a regional CTM, the Weather Research and Forecasting model coupled with Chemistry (WRF-Chem) (Grell *et al.*, 2005), to construct the "true" atmosphere. We then sample such a virtual atmosphere to retrieve synthetic GEMS and OMI ozone profiles based on optimal estimation (Rodgers, 2000) and to extract synthetic surface observations. Next, we use a global CTM GEOS-Chem (Bey *et al.*, 2001; Park *et al.*, 2004; Mao *et al.*, 2013) as the forward model for data assimilation to perform the *control run* and *assimilation run*. Last, we quantify the information contributed by the observing system by comparing paired differences between these two simulations and the "true" atmosphere.

### 2.1 Simulation models

We apply a regional CTM, WRF-Chem (version 4.1) (Grell *et al.*, 2005), for the *nature run* of the OSSE to simulate the "true" state of atmospheric compositions. The selection of *nature run* is crucial to characterize the realistic model errors dedicated to the robustness of the OSSE. Here we configure WRF-Chem with a grid resolution of 50 km × 50 km covering most of Asia and 34 vertical layers extending from the surface to 50 hPa. The setup of the WRF-Chem simulation, as well as the GEOS-Chem simulations described below, follows our preceding work (Shu *et al.*, 2022) and is summarized in Table S1. We have previously shown its robust capability in reproducing the temporal variations and vertical distributions of ozone through validation against actual surface ozone and ozonesonde measurements, respectively.

For the *control run* and *assimilation run*, we use a nested version of the GEOS-Chem (version 12.9.3; http://www.geos-chem.org, last access: 28 October 2022), which is a global CTM with a detailed $HO_x$-$NO_x$-VOC-ozone-aerosol-halogen





tropospheric chemistry mechanism (Bey *et al.*, 2001; Park *et al.*, 2004; Mao *et al.*, 2013) and has been widely applied in ozone simulations (Zoogman *et al.*, 2014; Gong *et al.*, 2020; Shu *et al.*, 2022). The nested Asian (11°S–55°N, 60°–150°E)

simulation has a resolution of 0.5° × 0.625°, with 47 vertical layers up to 0.1 hPa and boundary conditions updated every 3 h from a global 2° × 2.5° simulation. As indicated in Table S1, we configure GEOS-Chem as differently as possible in meteorological fields, chemical mechanisms, and emission inventories relative to WRF-Chem simulation to maximize their independence for the proper interpretation of OSSE results, as previously emphasized.

### 2.2 Observing system and synthetic observations

Our OSSE simulates the prospective ozone observing system over Asia during the operation of GEMS, consisting of GEMS geostationary satellite measurements, LEO OMI (henceforth LEO) satellite measurements, and surface measurements. We produce synthetic observations since GEMS scientific products have not yet been released. We assume the normal operation of OMI, acknowledging data loss due to row anomalies and instrument degradation (Schenkeveld *et al.*, 2017), which will be addressed in our future study assimilating real observations.


To simulate GEMS and OMI ozone retrievals, we apply a fast ozone profile retrieval simulation (FOR) tool (Shu *et al.*, 2022), in which the optimal estimation-based retrievals (Rodgers, 2000) and lookup table (LUT)-based radiative transfer simulations (Bak *et al.*, 2021) are merged. The spectral range is set to be 305-340 nm for GEMS while both Hartley band (269–309 nm) and Huggins band (312–330 nm) are used for OMI (Bak *et al.*, 2013; 2019). Hourly GEMS ozone profiles are

synthetically retrieved during the daytime (01-08 UTC) over East Asia (5°S–45°N, 75°–145°E). We likewise generate daily OMI measurements within the study domain according to the real-time orbit information of OMI (https://disc.gsfc.nasa.gov/datasets/OMTO3_003/summary, last access: 28 October 2022).

Briefly, the *a posteriori* estimate of $x$ ($\hat{x}$) is determined as a linear combination of the true state ($x_t$) and *a priori* ($x_{ap}$) state,

with **A,** weighting factor:

$$\hat{x} = \mathbf{A}x_t + (\mathbf{I} - \mathbf{A})x_{ap} + \mathbf{G}\hat{\varepsilon} \tag{1}$$

$$\mathbf{A} = \left(\mathbf{K}^T \mathbf{S}_y^{-1} \mathbf{K} + \mathbf{S}_a^{-1}\right)^{-1} \mathbf{K}^T \mathbf{S}_y^{-1} \mathbf{K} = \hat{\mathbf{S}} \mathbf{K}^T \mathbf{S}_y^{-1} \mathbf{K} = \mathbf{G}\mathbf{K} \tag{2}$$

where $\mathbf{S}_y$ and $\mathbf{S}_a$ are covariance matrics of measurement random-noise errors and *a priori* errors, respectively, with **K**, weighting function matrix ($\mathbf{K} \equiv \partial y / \partial x_t$). **A** is the averaging kernel matrix ($\mathbf{A} \equiv \partial \hat{x} / \partial x_t$) representing the sensitivity of the

retrieved profile to the true state (measurement information). Each diagonal element of **A** represents the degree of freedom for signal (DFS), quantifying the number of independent pieces of information available at that layer from radiance measurements. **I** is the identity matrix. To account for the impact of measurement noises on *a posteriori* estimate, $\mathbf{G}\hat{\varepsilon}$ is



added to the *a posteriori* estimate, where $\mathbf{G}$ describes the relative contribution of measurement errors ($\hat{\boldsymbol{\varepsilon}}$) on the retrieval ($\mathbf{G} \equiv \partial\hat{\boldsymbol{x}}/\partial\boldsymbol{y}$). $\hat{\mathbf{S}}$ is the covariance matrix of solution errors, which is also seen as the sum of $\mathbf{S}_n$ and $\mathbf{S}_s$:

$$\mathbf{S}_n = \mathbf{G}\mathbf{S}_y\mathbf{G}^T \tag{3}$$

$$\mathbf{S}_s = (\mathbf{A} - \mathbf{I})\mathbf{S}_a(\mathbf{A} - \mathbf{I})^T \tag{4}$$

where $\mathbf{S}_n$ and $\mathbf{S}_s$ quantify the retrieval errors caused by the measurement errors and *a priori* errors, respectively.

In this experiment, *a priori* information ($\boldsymbol{x}_{ap}, \mathbf{S}_a$) is taken from the climatological dataset (McPeters and Labow, 2012), commonly for GEMS and OMI data simulation. The state vector consists of the ozone profiles at 24 layers, surface albedo, and cloud fraction. The pressure level grid is set at $P_i = 2^{-\frac{i}{2}} atm$ for $i = 0$ to 23 (1 $atm$ = 1013.25 hPa, ~ 2.5 km-thickness between levels), with the top of the atmosphere set for $P_{24}$ (~ 55–65 km). To complete the "true" profiles, the WRF-Chem "true" atmosphere is used and extended with GEOS-Chem *control run* outputs above ~ 50 hPa due to the top pressure limit of WRF-Chem and then are spatially and temporally nested onto the standard retrieval grids. Both GEMS and OMI synthetic datasets are simulated at native spatial grids, and then the spatial binning is applied by 4 × 4 for GEMS and 4 (along-track) × 2 (across-track) for OMI to match with the planned GEMS data format and the current OMI data format (https://avdc.gsfc.nasa.gov/pub/data/satellite/Aura/OMI/V03/L2/OMPROFOZ, last access: 28 October 2022), respectively.

Figures 1 and 2 compare the performance of GEMS (Fig. 1e and f) and OMI (Fig. 2b and c) retrievals, including ozone profiles ($\hat{\boldsymbol{x}}$), associated retrieval errors ($\sqrt{\hat{\mathbf{S}}}$), and averaging kernels ($\mathbf{A}$) at noon for a specified location near Beijing with a high ozone level (Figs. 1b and 2a). In general, the magnitude of GEMS retrieval errors is very close to that of *a priori* errors at the upper atmospheric layers above ~ 2 hPa due to the weak retrieval sensitivity (DFS close to zero) and hence the strong influence of *a priori* on the retrievals (Bak *et al.*, 2013). In comparison to GEMS, the retrieval quality of OMI is significantly better above 200 hPa for representing the stratosphere and upper troposphere, with the DFS value up to 0.53. Nevertheless, GEMS has stronger vertical sensitivity and smaller retrieval errors relative to OMI at each layer below 200 hPa, providing sufficient measurement information to characterize tropospheric and near-surface ozone. Specifically, GEMS has more potential to capture the hot spots and diurnal evolution of ozone pollution in East Asia (Fig. 1a–d), as revealed in Shu *et al.* (2022).

We sample hourly surface ozone measurements based on site information from three networks, *i.e.*, the Tropospheric Ozone Assessment Report (TOAR) (Schultz *et al.*, 2017), the China National Environmental Monitoring Center (CNEMC) (http://www.cnemc.cn, last access: 28 October 2022), and the Continuous Ambient Air Quality Monitoring Stations (CAAQMS) of Central Pollution Control Board (CPCB) in India (Singh *et al.*, 2020). There are a total of 3,214 sites in the Asian domain (Fig. 3).



### 2.3 Assimilation of satellite and surface observations

We adopt a sequential sub-optimal Kalman filter technique (Parrington *et al.*, 2009; Zoogman *et al.*, 2011, 2014; Shu *et al.*, 2022) in the data assimilation system. At each assimilation time step, we calculate the optimal profile $\widehat{x}^a$ as a weighted average of the model forecast $x^b$ and the observation $\widehat{x}^{obs}$:

$$\widehat{x}^a = x^b + \mathbf{M}(\widehat{x}^{obs} - \mathbf{H}x^b) \tag{5}$$

where $\mathbf{H}$ represents the observation operator that maps the model forecast into the observation space. For satellite measurements $\mathbf{H}x^b = x_{ap} + \mathbf{A}(\mathbf{S}x^b - x_{ap})$, it utilizes the spatial interpolation operator $\mathbf{S}$, the *a priori* profile $x_{ap}$, and averaging kernels $\mathbf{A}$ from satellite retrievals to remove the dependence of the analysis on the model-retrieval comparison (Miyazaki *et al.*, 2012, 2020b). We limit our assimilation exercise to satellite pixels with cloud fraction less than 0.3 and vertical profiles at the bottom 11 retrieval layers (see Figs. 1 and 2) to avoid introducing redundant stratospheric information. For surface measurements, $\mathbf{H}x^b = x^b$.

The Kalman gain matrix $\mathbf{M}$ measures the relative weight apportioned to the model forecast and the observation:

$$\mathbf{M} = \mathbf{P}^b\mathbf{H}^T(\mathbf{H}\mathbf{P}^b\mathbf{H}^T + \mathbf{R})^{-1} \tag{6}$$

where $\mathbf{P}^b$ is the model error covariance matrix that expresses the errors in the forward model. Following Zoogman *et al.* (2014) and Shu *et al.* (2022), we use the square of 25 % of the background profile to initialize the model error variances, *i.e.*, the diagonal terms. We parameterize the model error covariances (off-diagonal terms) using the horizontal and vertical error correlation lengths of 450 km and 1.7 km, respectively (Fig. S2). We update this matrix at each assimilation time step by $\mathbf{P}^a = (\mathbf{I} - \mathbf{M}\mathbf{H})\mathbf{P}^b$, where $\mathbf{I}$ is the identity matrix.

$\mathbf{R}$ is the observation error covariance matrix, including the contributions from the measurement error and the representativeness error. Here $\mathbf{R}$ is assumed to be diagonal, that is, the observation errors are not correlated. For satellite measurements, the observation error (*i.e.*, solution error from Section 2.2) is determined during the retrieval procedure and reduced by the square root of the number of observations averaged over each GEOS-Chem grid square (Zoogman *et al.*, 2014; Shu *et al.*, 2022). We assume zero representation error since the synthetic satellite observations are spatially dense. For surface measurements, the measurement error $\varepsilon_0$ is assumed to be 4 % for ozone, according to officially released documents of the Chinese Ministry of Ecology and Environmental Protection (HJ 193–2013 and HJ 654–2013, available at http://www.cnemc.cn/jcgf/dqhj/, last access: 28 October 2022) following Kong *et al.* (2021). The representativeness error is parameterized as proposed by Elbern *et al.* (2007):

$$\varepsilon_r = \sqrt{\Delta x/L} \times \epsilon^{abs} \tag{7}$$



where $\Delta x$ is the model resolution ($\sim$ 56 km), $L$ represents the characteristic representativeness length of the surface stations,

and is set as 2 km due to the lack of observation site information. $\epsilon^{abs}$ represents the error characteristic parameters (1.2 for

ozone) according to Elbern *et al.* (2007). Finally, the total observation error is defined as:

$$\varepsilon_t = \sqrt{\varepsilon_0^2 + \varepsilon_r^2} \qquad\qquad (8)$$

Since the horizontal resolution of all synthetic observations (GEMS, LEO satellite, and surface observations) is much finer

than that of the model, we apply a super-observation approach to produce more representative data and reduce the horizontal

observation error correlations (Ma *et al.*, 2019). A super-observation is generated by averaging all observations located

within the same 0.5° latitude × 0.625° longitude GEOS-Chem model grid. Eventually, we reduce more than 3,000 surface

monitoring sites to 725 super-observation grids, and then randomly select 80 % of these super-observations for assimilation

and 20 % for validation, as shown in Fig. S3.

To distinguish the impact of assimilated observations and assimilation time step on the performance, we conduct eight data

assimilation experiments (Table 1). We assimilate daytime synthetic GEMS, LEO satellite, and surface observations at

successive 1 h or 3 h time steps individually or simultaneously. We use the mean bias error (MBE), mean absolute error

(MAE), root-mean-square error (RMSE), and correlation coefficient ( $r$ ) to assess the assimilation performance in

reproducing surface and tropospheric ozone relative to the "true" atmosphere.

## 3 Results and discussion

### 3.1 Improved simulations of surface ozone

Our analysis starts with evaluating the benefit of multiple ozone observations on surface ozone simulation. Figure 4

compares the surface MDA8 ozone concentrations and the frequency of high-ozone days (hereafter defined as surface

MDA8 ozone > 80 ppbv) in June 2020 between the "true" atmosphere and GEOS-Chem simulations without assimilation (*a*

*priori*) and with assimilation (*a posteriori*) of synthetic GEMS, surface, and LEO observations at 1 h time steps (Exp 1–4).

Compared with the "true" atmosphere, the GEOS-Chem *a priori* is biased low (–2.1 ppbv for regional mean bias) and

performs poorly in capturing excessive ozone. In the "true" atmosphere, the frequency of high-ozone days is found to be $\sim$

1.79 on average over the Asian domain, with recurrent ozone exceedance in eastern China, Tibet Plateau, and northern India.

However, the *a priori* only captures 0.2 high-ozone days per grid square per month with weak spatial correlation ($r$ = 0.51).

The individual or simultaneous assimilation of GEMS and surface observations effectively corrects the negative MDA8

ozone bias (dropping from 2.1 to 0.2–1.2 ppbv) and better captures ozone exceedance (increasing to 0.46–0.75 days). We

also observe an improved spatial pattern of simulated ozone against the "true" atmosphere, with $r$ increasing by 0.01–0.05

for MDA8 ozone concentrations and by 0.07–0.23 for high-ozone days.





On the whole, the added value of GEMS (Exp 1) to surface ozone simulations is smaller but non-negligible than that of surface observations (Exp 2), while the simultaneous assimilation of GEMS and surface data (Exp 3) provides the best performance among these *assimilation runs*. On that basis, we further discover that the addition of the LEO instrument (Exp

4) may not provide new valuable information to constrain surface ozone beyond GEMS and surface observations (Exp 3), as similarly indicated in Zoogman *et al.* (2014) because OMI and GEMS have similar spectral information on surface ozone.

To investigate the influence of assimilation frequency, we perform four sensitivity experiments with a longer assimilation time step of 3 h (Exp 5–8, see Fig. S4). We find that additional assimilation of LEO measurements (Exp 8) decreases the

mean bias from –0.8 to –0.4 ppbv and improves the prediction of high-ozone days from 0.49 to 0.57 days with a stronger spatial correlation ($r$ increases by 0.05) beyond GEMS and surface data (Exp 7). This highlights the importance of the LEO instrument in constraining surface ozone when observations from other platforms are inadequate for assimilation.

Considering the heterogeneous spatial corrections made to surface ozone after data assimilation in Asia, which is likely a

cause of the space layout of the observing system, we thereby review the statistics for 18 designated Asian countries (Fig. 5). We show the reduction in MAE and RMSE, and the differences in spatial correlation between GEOS-Chem simulations with and without data assimilation relative to the "true" atmosphere. We rule out the assimilation experiments involving the LEO instrument considering its insignificant impacts on surface ozone simulations, as discussed above.

Taking the *assimilation runs* with a 1 h time step (Exp 1–3) as an example, our analysis confirms that the joint assimilation of GEMS and surface data has the best performance in more than half of the Asian countries, with MAE reduced by 7–74 % (except for North Korea and Burma) and RMSE reduced by 5–69 % (except for Japan, North Korea, and Burma) in simulated ozone. In Southeast Asia, the improvements are almost spatially consistent with RMSE reduced by 17–42 % in most countries when concurrently assimilating GEMS and surface data, especially notably in Cambodia, Vietnam, and the

Philippines (by 31–42 %). Meanwhile, the data assimilation results including only GEMS observations (Exp 1) and with the addition of surface observations (Exp 3) have comparable performance in some regions (*e.g.*, the Philippines, Indonesia, and Vietnam), revealing the dominant role of GEMS rather than surface observations in improving surface ozone simulations as a result of the spatially sparse surface observations (Fig. 3). In comparison, we may relate the elevated benefit of surface observations beyond GEMS on ozone simulations in some countries, like Thailand and Laos, to the propagation of

information from more dense surface observations in East Asia through transboundary transport.

In East Asia, we observe a steady improvement in the spatial correlation of simulated ozone (*e.g.*, by up to 0.08 in China) relative to the "true" atmosphere after assimilating surface observations individually (Exp 2) or in combination with GEMS data (Exp 3). Specifically, the improvements in ozone simulations between these two simulations are quantitatively close in



China, with a reduction of ~ 32–35 % in MAE and RMSE. However, the influence of assimilation efforts is complicated in East Asia. In Japan and Mongolia, the assimilation of GEMS data generally contributes to a deterioration of simulated ozone and even counteracts the positive impact of surface observations when performing the joint assimilation, which may similarly be attributed to the influence of transboundary transport.

In South Asia, slight improvements in surface ozone simulations are achieved in India, Nepal, Bangladesh, and Sri Lanka after assimilating GEMS data due to the lack of observations (Fig. 1). Surface observations provide almost all the information to correct ozone bias when assimilating surface data individually (Exp 2) or with additional GEMS data (Exp 3), where we detect an MAE and RMSE reduction of 7–52 % and 5–49 %, respectively. Differently, GEMS could provide meaningful information to track ozone in Bhutan (RMSE reduced by over 70 %), located within its observational field. We neglect here the slightly-worsened spatial pattern of simulated ozone in Bhutan, mainly resulting from the spatially scattered and sparse model grids, as observed in Vietnam, Cambodia, and Malaysia.

We reach similar conclusions when expanding our analysis to the other *assimilation runs* with a longer time step of 3 h (Exp 5–7). In general, we find that the spatial coverage of multiple observations and the spatial spread of observational information via transboundary transport would significantly influence the assimilation performance, resulting in unequally distributed improvements in surface ozone simulations over Asia. The intercomparison of these two groups of experiments further shows that the assimilation of GEMS observations individually or in combination with surface data at 3 h time steps could achieve comparable performance in some regions (*e.g.*, Laos and Vietnam) relative to experiments assimilating all observations at 1 h time steps. This implies the feasibility of reducing the number of assimilated observations in actual applications to improve computational efficiency.

GEMS will provide continuous daytime measurements of tropospheric ozone profiles, thus the capability of geostationary observations through data assimilation to monitor the hourly variations of surface ozone is of particular interest. Figure 6 presents the diurnal variations of surface ozone at validation grids (Fig. S3). Overall, the joint assimilation of GEMS and surface observations (Exp 3) shows the best performance in reproducing the temporal variability of simulated ozone (Fig. 6a), with the smallest bias and RMSE in the daytime, especially in the late afternoon (Fig. 6b and c). Notably, it adds more valuable information to constrain ozone for the period of 03–11 (03–08) UTC than the individual assimilation of GEMS (surface) measurements, principally depending on the temporal coverage of these observations. Besides, the *a priori* shows an average temporal correlation coefficient ($r$) of 0.55 in simulated hourly ozone relative to the "true" atmosphere. Individually-assimilating GEMS data (Exp 1) makes limited corrections to this correlation, which is in sharp contrast to the enhanced temporal correlation from the data assimilation results including only surface data (Exp 2) and with additional GEMS data (Exp 3), improving the average of $r$ to 0.72 and 0.70, respectively (Fig. 6d).





To test the value of multiple ozone observations for short-term ozone forecasts, we conduct a series of 72 h forecasts, each
initialized at 06:00 UTC from 1 to 27 June 2020 (not shown). The 27 forecasting experiments are conducted using the
concentration analysis from Exp 3 as the chemical initial conditions. Our results suggest no substantial improvement in the
short-term ozone forecasts, as evidenced in Fig. 6c. Note that the adjustments in simulated ozone vanish rapidly after 11
UTC without new observations assimilated, leading to a spiked increase in ozone RMSE. We attribute this to the
unimplemented optimization of ozone precursors (Timmermans *et al.*, 2019). Further efforts concerning the simultaneous
optimization of the chemical initial conditions and ozone precursor emissions ($NO_x$ and VOCs) are essential to improve the
short-term ozone forecasts (Ma *et al.*, 2019; Peng *et al.*, 2018).

## 3.2 Improved simulations of tropospheric ozone profiles

Next, we evaluate to what extent tropospheric ozone profile simulations may benefit from the individual or simultaneous
assimilation of multiple ozone observations. Figure 7 displays the simulated ozone profiles averaged over the Asian domain
and three specified subregions, *i.e.*, East Asia, Southeast Asia, and South Asia (defined in Fig. 1). Figure 8 further compares
the MDA8 ozone concentrations at three vertical levels, *i.e.*, 200, 500, and 700 hPa, representing the upper, middle, and
lower troposphere, respectively. Here we focus on two *assimilation runs* using a 3 h time step (Exp 7 and 8) to pinpoint the
added value of the LEO instrument for tropospheric ozone simulations. See Figs. S5 and S6 for the *assimilation runs* using a
1 h time step (Exp 3 and 4).


Compared with the "true" atmosphere, the *a priori* tends to underestimate ozone from the surface to the upper troposphere
over all the specified regions (Fig. 7). The data assimilation greatly removes the negative bias and large RMSE, with more
obvious improvement in the middle to upper troposphere, especially in Southeast Asia. Data assimilation results without
(Exp 7) and with (Exp 8) the addition of the LEO instrument suggest that GEMS observations may have masked the added
value of LEO measurements for the whole Asian domain as well as East Asia and Southeast Asia, with a tiny discrepancy in
improving ozone vertical distributions. On the contrary, the LEO measurements add valuable corrections to the upper
tropospheric ozone simulations over South Asia where GEMS observations are unavailable. In addition, the changes in the
spatial correlation (varying from –0.05 to 0.04) of simulated ozone relative to the "true" atmosphere in East Asia and South
Asia are relatively small, showing a weakened spatial correlation in the middle to upper troposphere (above 600 hPa). On a
regional scale, in contrast, the data assimilation slightly increases the spatial correlation at the upper layers above ~ 350 hPa
over Asia. This improvement is predominantly due to the greatly improved ozone simulations in Southeast Asia in the
middle to upper troposphere, where an enhancement by up to ~ 0.16 in the spatial correlation is broadly distributed.

Vertically, our results show that the *a priori* is consistently biased low relative to the "true" atmosphere, with a regional
mean difference of –49.2, –15.1, and –6.3 ppbv at 200, 500, and 700 hPa, respectively. Jointly-assimilating GEMS and
surface observations (Exp 7) contributes to an apparent bias reduction of 6.6, 2.6, and 1.5 ppbv at the three levels,

respectively (Fig. 8). On that basis, the application of the LEO instrument (Exp 8) adds extra information to constrain ozone, with a superimposed bias reduction of 1.9, 0.7, and 0.5 ppbv, respectively. Particularly, the LEO measurements bring more practical corrections to ozone simulations in South Asia (*e.g.*, northern India at 500 hPa), as discussed above. In comparison,
we obtain a comparable performance (RMSE decreased by 7.2, 2.7, and 1.7 ppbv, respectively) from the joint assimilation of all GEMS and surface data (Exp 3) at 1 h times steps (Fig. S6), however, with no substantial corrections when further assimilating LEO data (Exp 4). This result is consistent with that for surface ozone simulations, that is, the application of all available GEMS observations may mask the direct added value of LEO data for tropospheric ozone simulations. Here we neglect the small changes in the spatial pattern of simulated ozone, given that satellite measurements do not fully cover the
Asian domain.

Figure 9 compares the RMSE reduction in simulated ozone at 200, 500, and 700 hPa for the six *assimilation runs* in 18 Asian countries. Here we rule out the experiments that individually assimilate surface data (Exp 2 and 6). At 200 hPa, our analysis reveals a robust improvement in ozone simulations for almost all the countries relative to the "true" atmosphere
from all assimilation experiments. We see an RMSE reduction of 33–57 % in most of Southeast Asia (except for Malaysia and Indonesia), 7–31 % in East Asia, and 4–81 % in South Asia. These results are generally in line with our previous work (Shu *et al.*, 2022), which reports an RMSE reduction of 18–49 % between 200–300 hPa after assimilating GEMS data individually, particularly with a better performance at low latitudes.

The performance of these six assimilation experiments is roughly comparable for East Asia and Southeast Asia, whereas the joint assimilation of synthetic observations at 3 h time steps (Exp 8) contributes to the most significant improvement (*e.g.*, RMSE decreased by > 80 % in Bangladesh) in ozone simulations in South Asia. This is also observed at 500 and 700 hPa, excluding the limited improvement in simulated ozone in South Asia after data assimilation except for Exp 8. We suggest the closer agreement in simulated ozone against the "true" atmosphere in South Asia that is only observed at 200 hPa is
primarily attributable to the larger retrieval sensitivities and smaller observation errors of satellite measurements (Figs. 1 and 2), as well as the spatial propagation of information as a result of the much more significantly improved tropospheric ozone simulations in Southeast Asia and East Asia.

At 500 hPa, the data assimilation contributes to an RMSE reduction of 30–37 % in Japan, South Korea, and North Korea,
which is more evident than that in China and Mongolia (9–25 %). In comparison, the improvements are much more unequally distributed in Southeast Asia. We see a higher RMSE reduction of up to 74 % in simulated ozone in the Philippines, followed by Thailand, Laos, Vietnam, and Cambodia (35–47 %). However, the assimilation also results in worse ozone simulations in regions like Malaysia and Indonesia.

At 700 hPa, there is a significant RMSE reduction of 37–69 % in most of Southeast Asia (except for Indonesia, the Philippines, and Burma), which is more prominent than that of East Asia (by 12–51 % except for North Korea). Moreover, it is important to note here that the additional assimilation of surface data (Exp 3) beyond GEMS (Exp 1) adds slightly visible corrections to ozone in some areas (*e.g.*, China, Cambodia, and Malaysia), while it offsets the positive effect of GEMS measurements and therefore leads to a weakened improvement in ozone simulations in regions like Mongolia, Laos, and

Vietnam. This illustrates the vertical propagation of information from surface observations to upper atmospheric layers.

**4 Conclusions**

We have applied two independent chemical transport models (GEOS-Chem and WRF-Chem) through an Observation System Simulation Experiment (OSSE) to investigate how the assimilation of multiple ozone observations (GEMS, LEO satellite, and surface observations), individually or in combination, would benefit the surface and tropospheric ozone

simulations in summer over Asia.

For surface ozone simulations, individual or joint assimilation of GEMS and surface observations reduces the simulation biases and improves the monitoring of ozone exceedance, spatial patterns, and diurnal variations. In most cases, the added value of GEMS observations for surface ozone simulations is smaller but non-negligible compared to surface data. The joint

assimilation of these two kinds of observations provides the best performance, with the mean bias reduced from –2.1 to –0.2 ppbv, the modeled high-ozone days increasing from 0.20 to 0.75 days, and the temporal correlation coefficient increasing from 0.55 to 0.70. However, these improvements in surface ozone simulations are unequally distributed over Asia, generally with a reduction of 5–69 % in the root-mean-square error (RMSE) in most countries. Specifically, we find that GEMS (surface) observations play a more critical role in constraining surface ozone in Southeast (South) Asia due to the sparse

distribution (absence) of surface (GEMS) measurements.

For tropospheric ozone profile simulations, the use of GEMS observations in data assimilation results in a more accurate prediction of ozone vertical distributions, especially in the middle to upper troposphere at low latitudes. On a regional scale, jointly-assimilating all GEMS and surface observations contributes to an apparent reduction of 7.2, 2.7, and 1.7 ppbv in the

mean ozone bias at 200, 500, and 700 hPa, respectively. Similar to those at the surface, we observe an inequality in the improvements in tropospheric ozone simulations. The data assimilation makes the most noticeable adjustments in ozone vertical distributions in Southeast Asia, exhibiting an enhanced spatial correlation of up to ~ 0.16 in the middle to upper troposphere and an RMSE reduction of 33–74 % at the three vertical levels in most countries. In contrast, the assimilation performance has the largest spatial variability in South Asia, with RMSE reduced by 4–81 % at 200 hPa but no improvement

achieved at the other two levels from most assimilation experiments.

The spatial coverage of assimilated observations, along with the horizontal and vertical propagation of information from multiple observations, significantly influences the assimilation performance, resulting in unequally distributed improvements in surface and tropospheric ozone simulations. In most cases, the join assimilation of synthetic observations at 1 h time steps

provides the best performance, whereas the assimilation experiments using 3 h time steps may have provided sufficient information in some cases, *e.g.*, to constrain the surface ozone simulations in regions of Southeast Asia and the upper tropospheric ozone simulations in South Asia. Besides, sensitivity experiments also reveal the enhanced role of the LEO measurements in improving surface and upper tropospheric ozone simulations, only when inadequate GEMS and surface observations are applied in data assimilation, especially in South Asia due to the absence of GEMS observations.


The improvements introduced by the multisource data assimilation using GEMS geostationary observations are promising, although the data assimilation experiments still have difficulty in fully reproducing the observed ozone features. As an extension of our preceding work (Shu *et al.*, 2022), this study offers a comprehensive simulation reference for future ozone studies in Asia, acknowledging that the improvement in ozone simulations should be interpreted within the current OSSE

framework. Further applications need to pay more attention to multiconstituent data assimilation to simultaneously optimize ozone and its precursor fields, including chemical initial conditions and emissions (Miyazaki *et al.*, 2012, 2020b; Ma *et al.*, 2019).

**Author contributions**

LZ devised the conceptual ideas and supervised the findings of this work. LS provided and processed the data, conducted the

investigation, performed formal analysis, and created the figures. LS, JB, and PZ developed the methodology. LS and JB contributed to the software. LZ, LS, and JB acquired the funding. LS drafted the original manuscript. LZ, JB, PZ, HH, SL, XL, SS, JL, YC, DP, XZ, WF, XY, and TMF reviewed and commented on the manuscript.

**Competing interests**

The authors declare that they have no conflict of interest.

**Acknowledgments**

This work is funded by the Key-Area Research and Development Program of Guangdong Province (2020B1111360001), Guangdong Basic and Applied Basic Research Foundation (2021A1515110713), Guangdong Basic and Applied Basic Research Fund (2020B1515130003), Guangdong University Research Project Science Team (2021KCXTD004), Shenzhen Science and Technology Program (KQTD20210811090048025, JCYJ20210324104604012, JCYJ20220530115404009), and



China Postdoctoral Science Foundation (2021M701554). This work is supported by the Center for Computational Science and Engineering at Southern University of Science and Technology. Research at Pusan National University is supported by the Basic Science Research Program through the National Research Foundation of Korea (NRF) funded by the Ministry of Education (2020R1A6A1A03044834).

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


## Figures and Tables

680 **Table 1. Configuration of data assimilation experiments.**

| No. | Experiments | Assimilated observations | | | Time step |
|---|---|---|---|---|---|
| | | **GEMS** | **Surface** | **LEO**[a] | |
| Exp 1 | GEMS 1h | √ | | | |
| Exp 2 | Surface 1h | | √ | | |
| Exp 3 | GEMS+Surface 1h | √ | √ | | 1 h |
| Exp 4 | GEMS+Surface+LEO 1h | √ | √ | √ | |
| Exp 5 | GEMS 3h | √ | | | |
| Exp 6 | Surface 3h | | √ | | |
| Exp 7 | GEMS+Surface 3h | √ | √ | | 3 h |
| Exp 8 | GEMS+Surface+LEO 3h | √ | √ | √ | |

[a]LEO observations are assimilated when available.

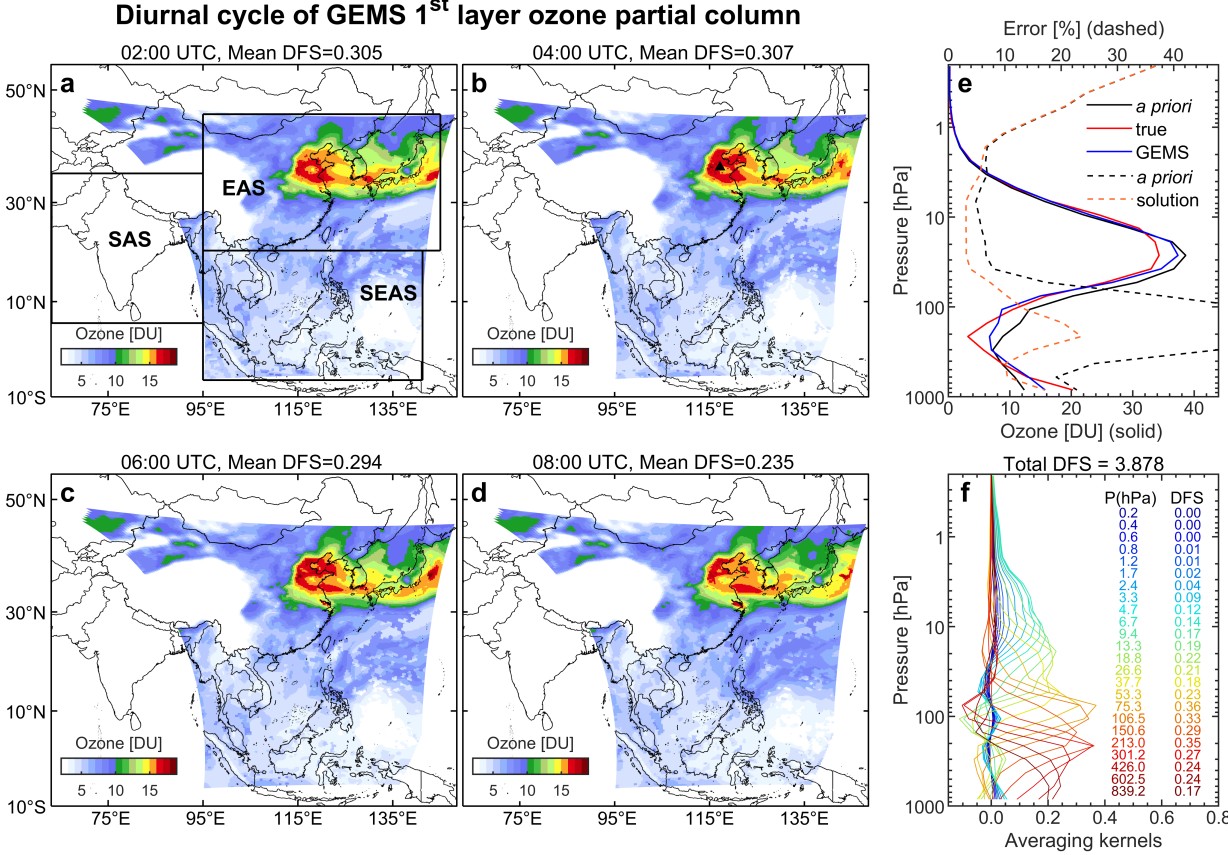

**Figure 1. Diurnal cycles of surface-layer ozone partial column (02:00, 04:00, 06:00, and 08:00 UTC, a–d), vertical ozone profiles and relative retrieval errors (e), and averaging kernels (f) at 04:00 UTC for a specified grid with high ozone level (36.6°N, 117.1°E, denoted using the black triangle in panel (b)) from GEMS retrievals on 16 June 2020. In panels (a–d), captions give the regional average of the degree of freedom for signal (DFS; defined as the trace of the averaging kernel matrix in Section 2.2) of the corresponding GEMS retrievals. The black box areas in panel (a) define the regions of East Asia (EAS), Southeast Asia (SEAS), and South Asia (SAS). In panel (e), the solid lines denote the *a priori* (black), true (red), and retrieved (blue) profiles. The dashed lines represent the *a priori* (black) and solution (orange) errors that both normalized to *a priori* profiles. In panel (f), the caption gives the total DFS. Averaging kernels (colored by layers) are normalized to the thickness of each layer and *a priori* errors. Also inserted are elements of the DFS vector along with the central pressure of each layer.**





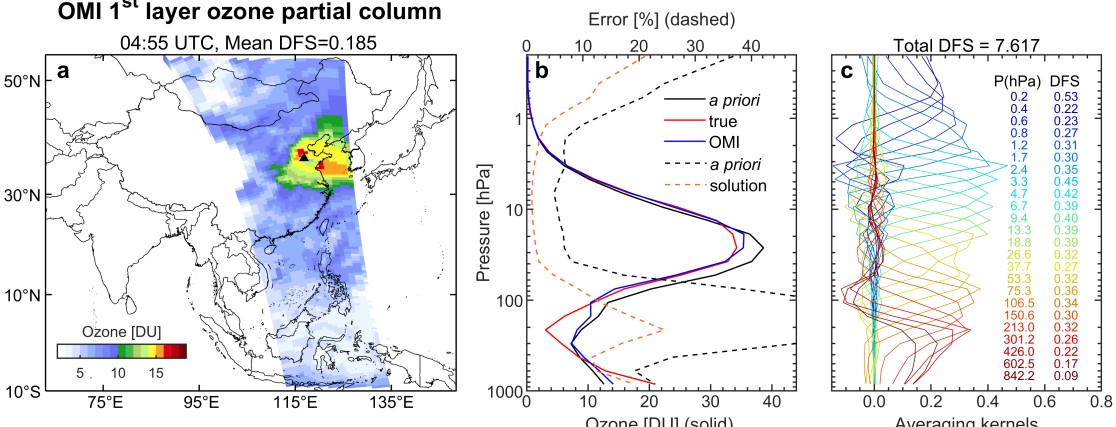

**Figure 2. Same as Fig. 1 but from OMI retrievals at 04:55 UTC on 16 June 2020.**

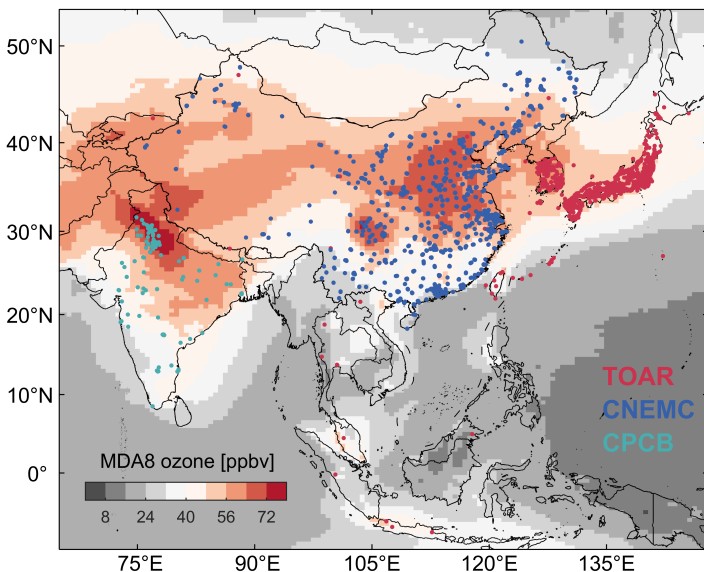

**Figure 3. Distributions of surface monitoring sites over the Asian domain. The red, blue, and green dots represent monitoring sites from the Tropospheric Ozone Assessment Report (TOAR), the China National Environmental Monitoring Center (CNEMC), and the Continuous Ambient Air Quality Monitoring Stations (CAAQMS) of Central Pollution Control Board (CPCB) in India, respectively. The colored areas indicate monthly mean daily maximum 8-h average (MDA8) ozone concentrations in June 2020, as simulated by the *control run*.**



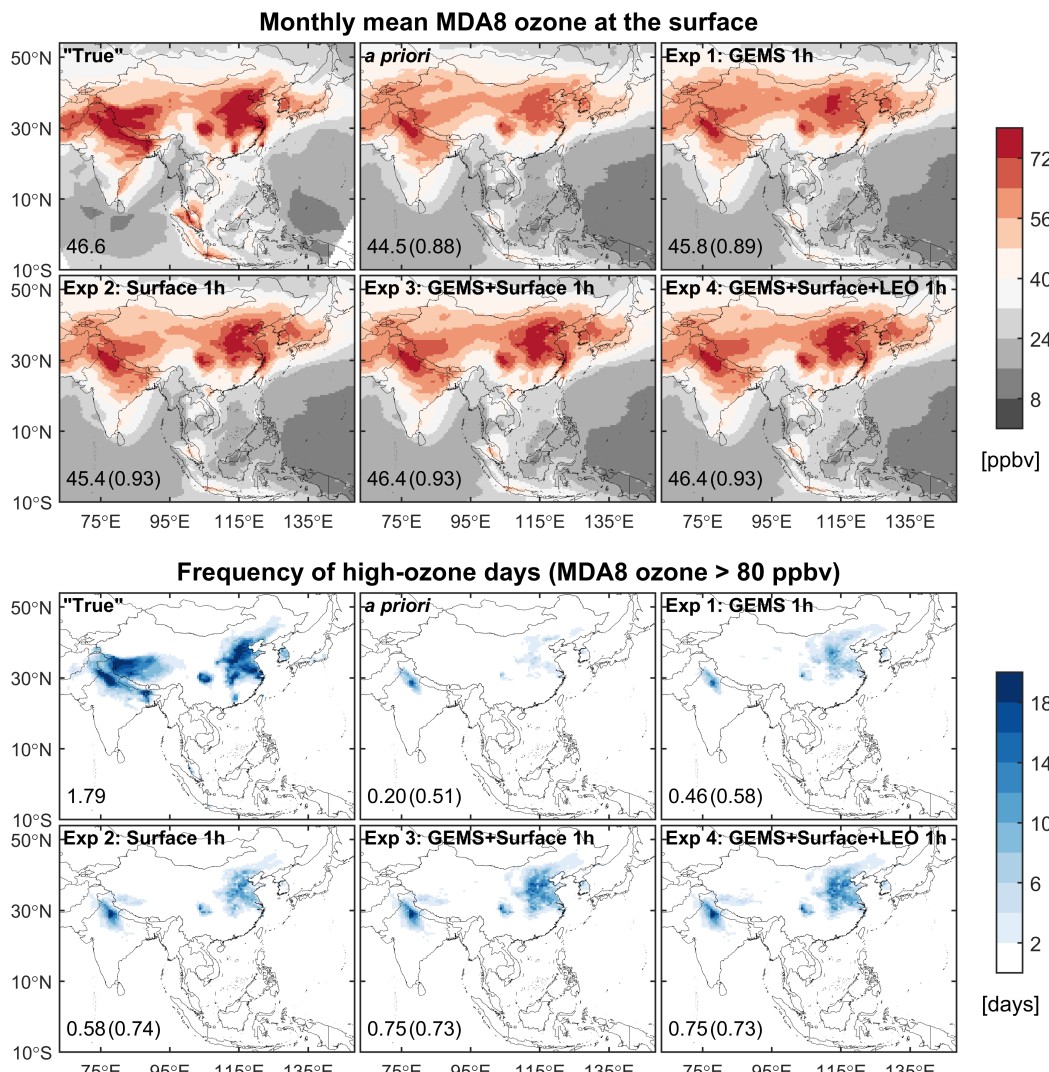

**Figure 4. Monthly mean daily maximum 8-h average (MDA8) ozone at the surface and the frequency of high-ozone days (defined as surface MDA8 ozone exceeding 80 ppbv) for June 2020 over the Asian domain, as simulated by the *nature run* ("True"), the *control run* (*a priori*), and four *assimilation runs* (Exp 1–4 in Table 1). The domain-averaged value over the land is inset. For the *control run* and *assimilation runs*, the spatial correlation coefficients relative to the *nature run* are also inset in parentheses.**

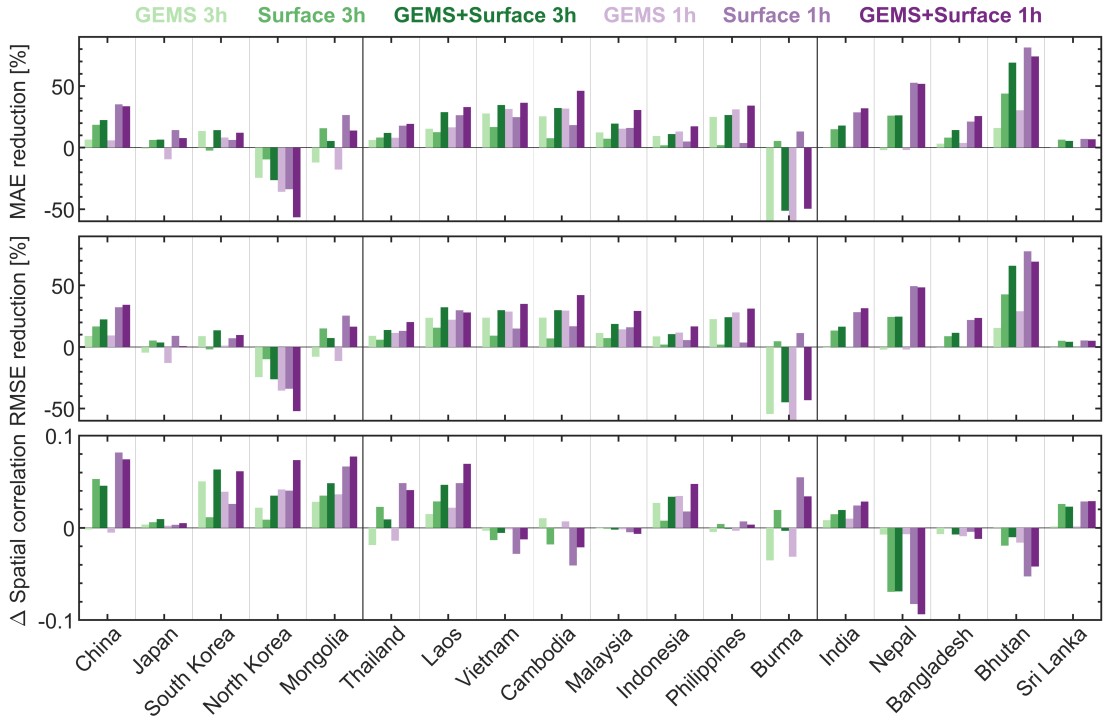

**Figure 5. The MAE reduction (top panel), RMSE reduction (middle panel), and the difference in spatial correlation coefficient (bottom panel) of simulated ozone between the *assimilation run* and the *control run* relative to the *nature run* for June 2020 in 18 Asian countries.**





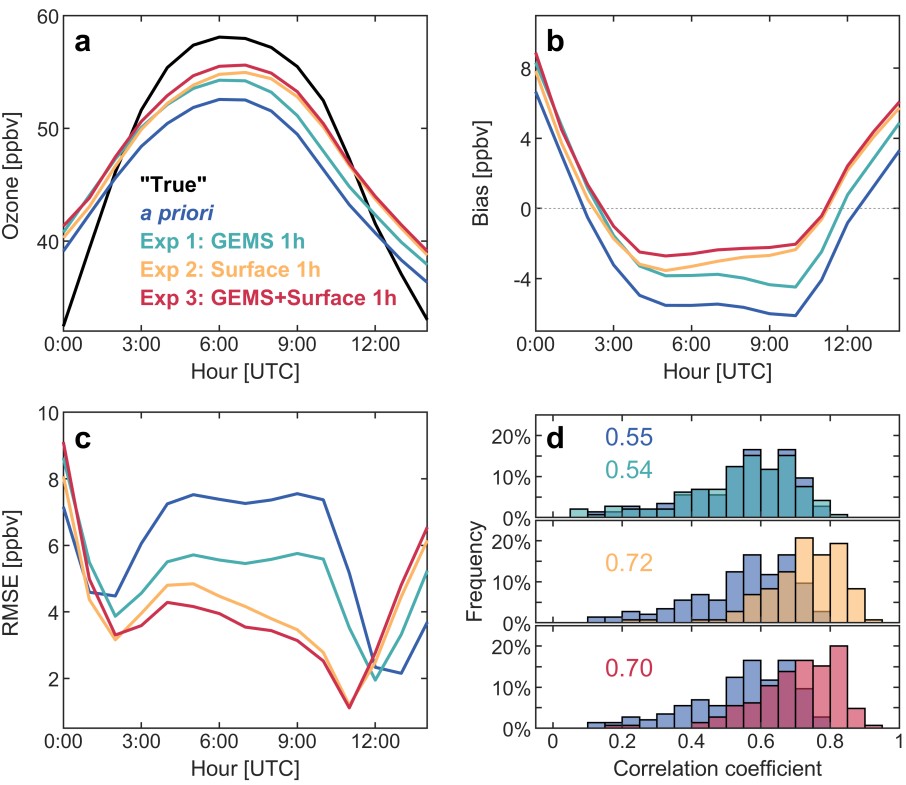

**Figure 6.** Averaged diurnal cycle of surface ozone (a), mean bias (b), RMSE (c), and the histogram of temporal correlation coefficient of hourly ozone (d) at validation grids (Fig. S3), as simulated by the *control run* (*a priori*) and three *assimilation runs* (Exp 1–3 in Table 1) relative to the *nature run* ("True") for June 2020. In panel (d), the mean values are inset.



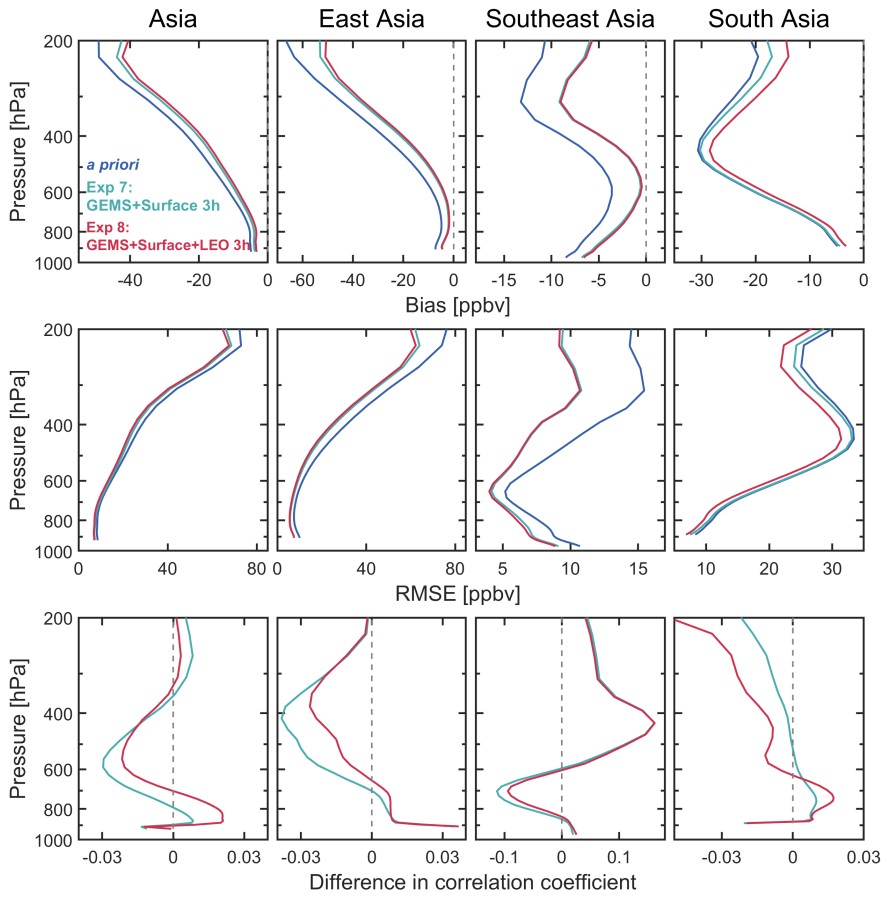

**Figure 7. Ozone MBE (top panels), RMSE (middle panels), and the difference in spatial correlation coefficient (*assimilation run* minus *control run*, bottom panels) relative to the *nature run* for June 2020 over the Asian domain, East Asia, Southeast Asia, and South Asia (defined in Fig. 1), as simulated by the *control run* (*a priori*) and two *assimilation runs* (Exp 7 and 8 in Table 1).**

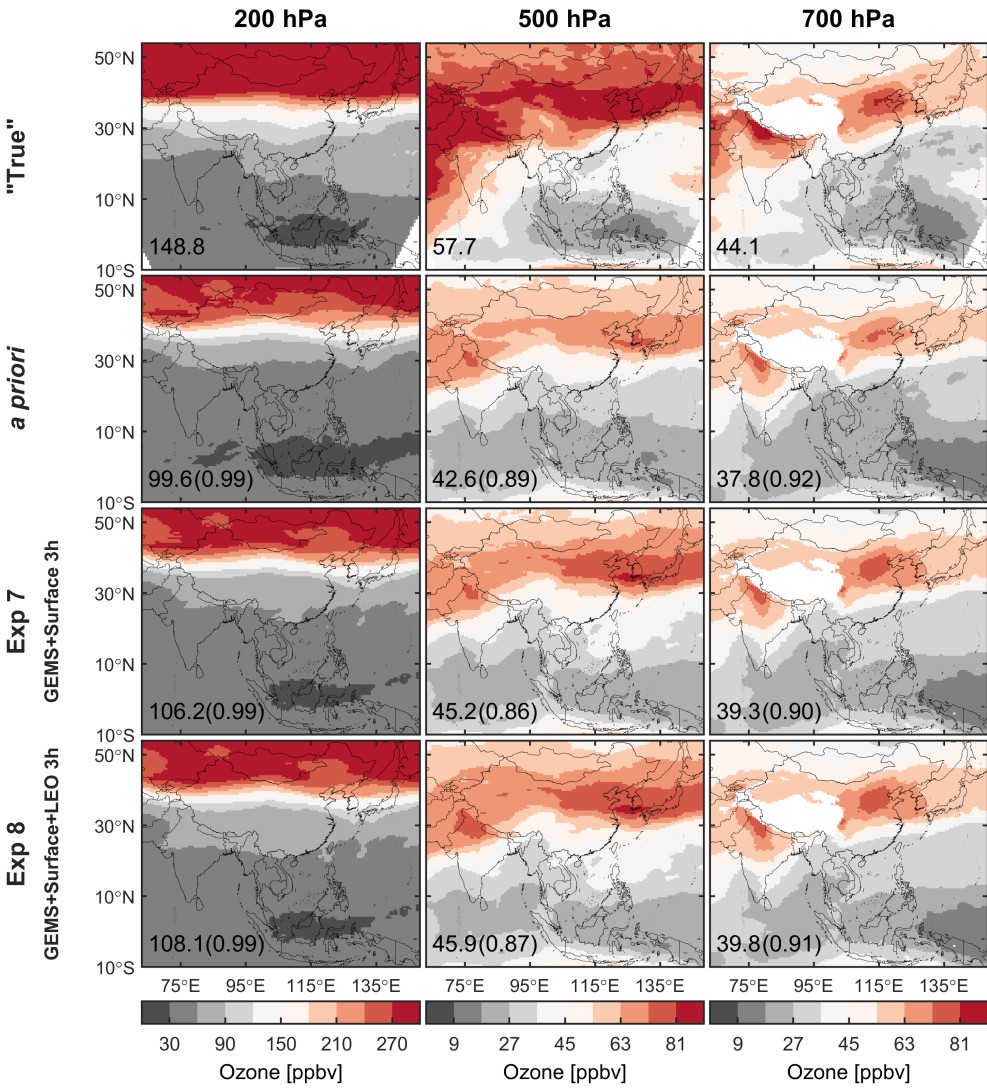

**Figure 8.** Monthly mean MDA8 ozone at 200 (left panels), 500 (middle panels), and 700 (right panels) hPa for June 2020 over the Asian domain, as simulated by the *nature run* ("True"), the *control run* (*a priori*), and two *assimilation runs* (Exp 7 and 8 in Table 1). The domain-averaged value is inset. For the *control run* and *assimilation runs*, the spatial correlation coefficients relative to the *nature run* are also inset in parentheses.





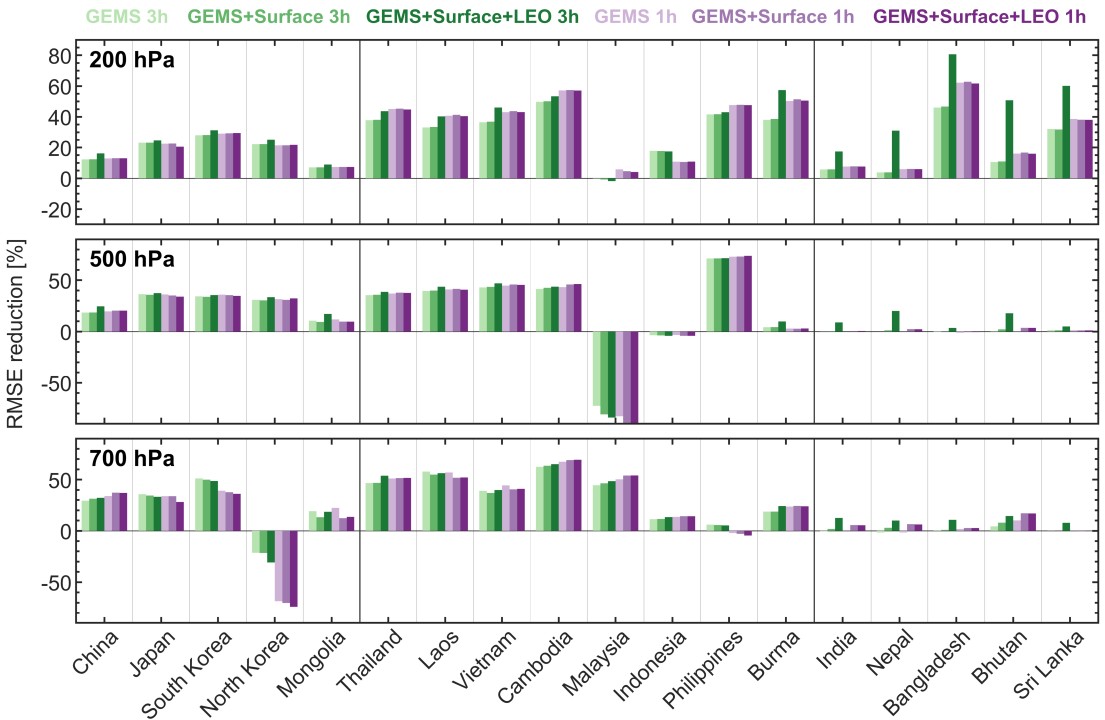

**Figure 9.** The RMSE reduction in simulated ozone between the *assimilation run* and the *control run* relative to the *nature run* for June 2020 in 18 Asian countries.