# Peer review of "Improving Ozone Simulations in Asia via Multisource Data Assimilation: Results from an Observing System Simulation Experiment with GEMS Geostationary Satellite Observations"

_Atmospheric Chemistry and Physics, 2022_

## Author Comment (AC1)

We thank the reviewer for the thoughtful comments. Our point-by-point responses to the comments are in blue. We have also carefully considered and addressed each comment in the revised manuscript.

**General Comments:**

The authors provided an OSSE-based analysis to investigate the impacts of satellite and surface O3 observations on the assimilated O3 concentrations over Asia. They found that joint assimilation by assimilating both satellite and surface O3 observations has the best performance. I agree with the authors that joint assimilation is helpful. However, the added value to the assimilated surface O3 by assimilating satellite observations is expected to be limited over areas with a high density of surface observations because of weaker sensitivity to surface O3 and larger observation errors in satellite observations. Additional analysis is suggested to demonstrate the difference between areas with and without a high density of surface observations, as it can clarify the advantage of satellite observations with respect to surface O3 observations. I recommend the paper for publication after consideration of the points below.

Response: Accepted.

In Fig. 5, we evaluate the added value of GEMS and surface observations for improving surface ozone simulations in 18 Asian countries. Our results have demonstrated the different assimilation performances in East Asia (with dense surface observations, see Fig. 3 for the distribution of surface monitoring sites), Southeast Asia (with sparse, limited surface observations), and South Asia (without GEMS observations, see Fig. 1).

[Figure]

**Figure 5.** The MAE reduction (a), RMSE reduction (b), and the difference in the spatial correlation coefficient (c) of simulated surface ozone between the *assimilation run* and the *control run* relative to the *nature run* for June 2020 in 18 Asian countries.

Per your suggestion (same as the 4th special comment), we have provided the additional analysis for eastern China (20–42°N, 110–123°E) with a higher density of surface observations (see Fig. S7 in the revised supplementary).

Generally, we similarly conclude that the added value of GEMS data to simulated surface ozone in eastern China is smaller but non-negligible than that of surface observations, while the joint assimilation of GEMS and surface data provides the best performance. This finding is generally consistent with that of China. More specifically, the joint assimilation of both GEMS and surface observations at 1 h time steps (Exp 3, in darkest purple, Fig. S7) contributes to an MAE (RMSE) reduction of 52 % (46 %) in simulated surface ozone in eastern China, which is more significant than those of China (both RMSE and MAE reduced by ~ 34 %). Besides, the improvements in the spatial pattern of simulated surface ozone in eastern China are limited (spatial correlation coefficient increased by ~ 0.02) and predominantly contributed by the assimilation of surface observations due to the smaller observation errors of surface observations compared to satellite observations.

We have revised this part in the revised Section 3.1 (lines 295–299) as follows:

"Specifically, in China, the improvements in ozone simulations between these two simulations are quantitatively close, with a reduction of ~ 32–35 % in MAE and RMSE. In comparison, here we extend the investigation to eastern China with a higher density of surface observations (Fig. S7) and observe that the joint assimilation has the best performance for improving surface ozone simulations, contributing to a more significant reduction of 52 % and 46 % in MAE and RMSE, respectively."

[Figure]

**Figure S7.** Same as Fig. 5 but for eastern China (20–42°N, 110–123°E).

**Special Comments:**

Lines 141-147: I understand that the fast ozone profile retrieval simulation (FOR) is necessary for GEMS because GEMS scientific products have not been released. I

suggest more analysis to demonstrate the consistency between FOR and OMI such as their averaging kernels and observation errors, as the conclusion of this work is based on that FOR is good enough to simulate satellite observations.

Response: Accepted.

In Fig. S2 of the revised supplementary, we have provided the validation of retrieval errors and averaging kernels of the OMI ozone profile at the Shanghai pixel (31.1°N 121.3°E) retrieved by the fast ozone profile retrieval simulation (FOR) tool against the Smithsonian Astrophysical Observatory (SAO) OMI Ozone Profile (PROFOZ) product (https://avdc.gsfc.nasa.gov/pub/data/satellite/Aura/OMI/V03/L2/OMPROFOZ/, last access: 01 March 2023). Considering the data loss due to row anomalies and instrument degradation of OMI (as seen in Fig. S2d), we perform the validation at a new location in Shanghai rather than the specified location near Beijing used in Figs. 1 and 2.

From Fig. S2, we learn that the retrieval errors are comparable in the troposphere and stratosphere from these two kinds of retrievals (Fig. S2b and e). However, the retrievals from the SAO PROFOZ product have higher solution errors near the surface and smaller solution errors above ~ 1 hPa relative to the FOR-based retrievals, which are largely associated with the *a priori* information.

In the SAO PROFOZ product, the ozone profile retrievals are constrained using the latitude-dependent *a priori* ozone profile climatology by McPeters *et al.* (2007) and OMI random-noise errors (https://avdc.gsfc.nasa.gov/pub/data/satellite/Aura/OMI/V03/L2/OMPROFOZ/OMPR OFOZ_readme-v3.pdf, last access: 01 March 2023). In comparison, the FOR tool uses *a priori* information ( $x_{ap}$ , $S_a$ ) from an updated ozone climatology (McPeters and Labow, 2012), commonly for GEMS and OMI data simulation. As for the averaging kernels, the comparison illustrates a good consistency in the vertical structure of retrieval sensitivity between these two kinds of retrievals (Fig. S2c and f), while the FOR-based retrievals generally have slightly higher retrieval sensitivity than those from the SAO PROFOZ product.

Overall, it could demonstrate the robust capability of the FOR tool in simulating satellite observations, acknowledging the slight differences in simulated retrieval errors and averaging kernels.

We have revised this part in the revised Section 2.2 (lines 183–186) as follows:

"In addition, the validation of retrieval errors and averaging kernels of the OMI ozone profile at Shanghai (Fig. S2) retrieved by the FOR tool against the Smithsonian

Astrophysical Observatory (SAO) OMI Ozone Profile (PROFOZ) product (https://avdc.gsfc.nasa.gov/pub/data/satellite/Aura/OMI/V03/L2/OMPROFOZ/, last access: 1 March 2023) demonstrate the robust capability of the FOR tool in simulating satellite observations."

[Figure]

**Figure S2.** Comparison of surface-layer ozone partial column (left panels), vertical ozone profiles and relative retrieval errors (middle panels), and averaging kernels (right panels) at the Shanghai pixel (31.1°N 121.3°E, denoted using the black triangle in left panels) at 04:55 UTC on 16 June 2020 between the OMI retrievals simulated by the fast ozone profile retrieval simulation (FOR) tool (a–c) and from the Smithsonian Astrophysical Observatory (SAO) OMI Ozone Profile (PROFOZ) product (d–f, (https://avdc.gsfc.nasa.gov/pub/data/satellite/Aura/OMI/V03/L2/OMPROFOZ/, last access: 1 March 2023). In panels (b) and (e), the solid lines denote the *a priori* (black), true (red), and retrieved (blue) profiles. The dashed lines represent the *a priori* (black) and solution (orange) errors that both normalized to *a priori* profiles. In panels (c) and (f), the caption gives the total degree of freedom for the signal (DFS; defined as the trace of the averaging kernel matrix). Averaging kernels (colored by layers) are normalized to the thickness of each layer and *a priori* errors. Also inserted are elements of the DFS vector along with the central pressure of each layer.

Figure 1 and Figure 2: I assume they are simulated retrievals rather than GEMS and OMI retrievals. In addition, the sensitivity of the 839 hPa level (GEMS, Figure 1f) is

uniform from the surface to 600 hPa; the sensitivity of the 842 hPa level (OMI, Figure 2c) is uniform from the surface to 300 hPa. Consequently, the contributions of assimilating GEMS and OMI on surface O3 concentration are expected to be limited with respect to surface O3 observations.

Response: Accepted.

We have revised the captions of Figs. 1 and 2 in the revised manuscript to clarify that we used the simulated satellite retrievals.

In Figs. 1 and 2, the vertical retrieval sensitivity to surface ozone at the specified location near Beijing is relatively uniform from the surface to ~ 600 hPa (~ 300 hPa) from simulated GEMS (OMI) retrievals. However, the averaging kernels are not uniform for different locations and different times, as shown in Fig. S2c (OMI retrievals at the Shanghai pixel) of the revised supplementary. A distinct feature of our work is the use of space- and time-varying averaging kernels (Shu *et al.*, 2022), making it different from air quality OSSEs reviewed by Timmermans *et al.* (2015). As a result, the added value to simulated ozone by assimilating GEMS data could be limited or promising in different regions over Asia relative to the assimilation of surface observations.

Lines 254-256: Can the authors perform a new experiment by only assimilating OMI? I am curious about the improvement which we can obtain by assimilating GEMS instead of OMI.

Response: Accepted.

We perform a new experiment by individually assimilating OMI data at 3 h time steps (denoted as Exp 9 in the revised Table 1).

Figure S6 in the revised supplementary shows the improvements in surface ozone simulations by individually assimilating LEO observations. We have discussed the difference in the assimilation performance between Exp 9 and Exp 5 (only assimilating GEMS observations, see Fig. S5) in the revised Section 3.1 (lines 266–273) as follows:

"Furthermore, we conduct an additional experiment that only assimilates LEO measurements (Exp 9, Fig. S6). Compared to LEO satellite observations, the individual assimilation of GEMS observations (Exp 5, Fig. S5) provides more added value for monitoring surface ozone in East Asia and Southeast Asia due to the stronger sensitivity to surface ozone and smaller retrieval errors (Figs. 1 and 2), especially more effectively diagnosing the ozone exceedance in eastern China, while makes little corrections to simulated surface ozone in South Asia owing to the limited coverage over this region (Fig. 1). However, this inability of GEMS observations to

correct ozone bias in South Asia could be addressed by additionally assimilating surface observations (Exp 7, Fig. S5). As such, the use of joint assimilation is essential to efficiently enhance the information of GEMS and surface observations to constrain surface ozone simulations."

[Figure]

**Figure S6.** Same as Fig. 4 but for the *assimilation run* Exp 9 (Table 1).

[Figure]

**Figure S5.** Same as Fig. 4 but for the *assimilation runs* with the assimilation time step of 3 h (Exp 5–8 in Table 1).

Figure S11 in the revised supplementary compares the improvements in tropospheric ozone profile simulations by individually assimilating GEMS (Exp 5) and LEO (Exp 9) observations. Vertically, the assimilation of GEMS observations also contributes to a more apparent bias and RMSE reduction in East Asia and Southeast Asia compared to LEO observations, especially notably in Southeast Asia in the middle to upper troposphere. In contrast, LEO satellite data could provide extra information to constrain ozone simulations in South Asia. In addition, the improvements in spatial correlation introduced by LEO satellite observations are relatively small and thus negligible due to its limited temporal coverage.

We have revised this part in the revised Section 3.2 (lines 354–361) as follows:

"Data assimilation results without (Exp 7) and with (Exp 8) the addition of the LEO instrument suggest that GEMS observations may have masked the added value of LEO measurements for the whole Asian domain as well as East Asia and Southeast Asia, with a tiny discrepancy in improving ozone vertical distributions. This is also demonstrated by the comparison of tropospheric ozone profiles between two *assimilation runs* that only assimilate GEMS or LEO observations (Exp 5 and 9, Fig. S11), where we see that the individual assimilation of GEMS data contributes to a more apparent bias and RMSE reduction in East Asia and Southeast Asia. On the contrary, the LEO measurements add valuable corrections to the upper tropospheric ozone simulations over South Asia (Fig. 7) where GEMS observations are unavailable (Fig. 1)."

[Figure]

**Figure S11.** Same as Fig. 7 but for the *assimilation runs* Exp 5 and 9 (Table 1).

Figure 5: What is the major added value of assimilating satellite measurements over areas with a high density of surface stations, such as E. China? I noticed that the spatial correlation in China is almost ZERO by assimilating GEMS and is about 0.55 by assimilating surface observations.

Response: Accepted.

Fig. 5c in the revised manuscript shows the difference in the spatial correlation coefficient between the *assimilation run* and the *control run* relative to the *nature run*. Here it needs to be clarified that assimilating surface observations at 3 h time steps (in green) contributes to an enhancement of ~ 0.053 (rather than ~ 0.55 as mentioned in the comment) in the spatial correlation coefficient of simulated surface ozone in China, while assimilating GEMS data makes no significant corrections to this spatial correlation.

Per your suggestion, we have provided the additional analysis for eastern China (20–42°N, 110–123°E) with a higher density of surface observations (see Fig. S7 in the revised supplementary). Please also see our response to the general comment.

Figure 6: Please provide more description for panel d. There are different colors and numbers shown in this panel and I don't understand what they represent. In addition,

because only one-month assimilation is performed in this work, it could be helpful to show the time series of daily or hourly O3 concentrations. It can better demonstrate the effect of various assimilations on various temporal scales.

Response: Corrected.

We have revised the caption of Fig. 6 (lines 729–736) to make it clear as follows:

"Figure 6. Comparison of the averaged diurnal cycle of (a) surface ozone, (b) mean bias, and (c) RMSE, as well as (d) the histogram (in percentage) of the temporal correlation coefficient of hourly ozone at validation grids (Fig. S4), as simulated by the *control run* (*a priori*, in blue) and three *assimilation runs* (Exp 1–3 in Table 1, in green, yellow, and red, respectively) relative to the *nature run* ("True", black line in panel a) for June 2020. In panel (d), the green, yellow, and red bars respectively represent the frequency (%) of the temporal correlation coefficient of simulated ozone between the three *assimilation runs* (Exp 1–3) and the *nature run*, which is in comparison to that of the *control run* (in blue). The blue bars are the same in the three sub-panels to better illustrate the improvements in the temporal correlation of simulated ozone relative to the *control run*. The mean values of the temporal correlation coefficient (colored the same as lines in panel a) at all validation grids are inset."

We agree that the time series of daily or hourly ozone concentrations could be helpful to better demonstrate the effect of various assimilations on various temporal scales. Currently, panels (a–c), and (d) of Fig. 6 have revealed the benefit of various observations on simulated surface ozone on the diurnal and hourly time scale, respectively. As we discussed in Section 3.1 as "GEMS will provide continuous daytime measurements of tropospheric ozone profiles, thus the capability of geostationary observations through data assimilation to monitor the hourly variations of surface ozone is of particular interest", thus we focus on the improvements on daytime hourly ozone simulations by assimilating various observations. In our experiments, we only assimilate daytime synthetic observations individually or simultaneously (described in Section 2.3). In addition, the unimplemented optimization of ozone precursors ($NO_x$ and VOCs) is a limitation of this study (discussed in Section 3.1). In this case, we observe no substantial improvements in simulated ozone in the nighttime without new observations assimilated, as well as in the short-term ozone forecasts (see the last paragraph of Section 3.1). As such, the comparison of the time series of hourly ozone concentrations is not provided because it cannot clearly show the added value to simulated ozone by assimilating various observations. Instead, the comparison of diurnal cycles of simulated ozone could demonstrate the improvements in surface ozone simulations on the hourly scale and also well illustrate the influence of temporal coverage of observations on assimilation performance.

Per your suggestion, we have added the comparison of the time series of daily MDA8 ozone concentrations (see Fig. S8 in the revised supplementary). We have revised this part in the revised Section 3.1 (lines 323–327) as follows:

"Figures 6 and S8 present the diurnal and daily variations of surface ozone at validation grids (Fig. S4), respectively. Overall, the joint assimilation of GEMS and surface observations (Exp 3) shows the best performance in reproducing the temporal variability of simulated surface ozone (Figs. 6a and S8a), with the smallest bias and RMSE, especially in the late afternoon (Fig. 6b and c) and on high-ozone days (Fig. S8b and c)."

[Figure]

**Figure S8.** Same as Fig. 6a–c but for the comparison of the daily variations.

**Reference:**

McPeters, R. D., Labow, G. J., and Logan, J. A.: Ozone climatological profiles for satellite retrieval algorithms, J. Geophys. Res. Atmos., 112, D05308, https://doi.org/10.1029/2005JD006823, 2007.

McPeters, R. D., and Labow, G. J.: Climatology 2011: An MLS and sonde derived ozone climatology for satellite retrieval algorithms, J. Geophys. Res. Atmos., 117, https://doi.org/10.1029/2011JD017006, 2012.

Shu, L., Zhu, L., Bak, J., Zoogman, P., Han, H., Long, X., Bai, B., Liu, S., Wang, D., Sun, W., Pu, D., Chen, Y., Li, X., Sun, S., Li, J., Zuo, X., Yang, X., and Fu, T.-M.: Improved ozone simulation in East Asia via assimilating observations from the first geostationary air-quality monitoring satellite: Insights from an Observing System Simulation Experiment, Atmos. Environ., 274, 119003, https://doi.org/10.1016/j.atmosenv.2022.119003, 2022.

Timmermans, R. M. A., Lahoz, W. A., Attié, J. L., Peuch, V. H., Curier, R. L., Edwards, D. P., Eskes, H. J., and Builtjes, P. J. H.: Observing System Simulation

Experiments for air quality, Atmos. Environ., 115, 199–213, https://doi.org/10.1016/j.atmosenv.2015.05.032, 2015.

---

## Author Comment (AC2)

The paper presents the OSSE experiments results to demonstrate the benefit of GEMS ozone observations in future applications. Both the methods and the data assimilation results with additional OMI and surface synthetic observations are well presented. However, improvement can be made if the authors can address the following concerns.

We thank the reviewer for the thoughtful comments. Our point-by-point responses to the comments are in blue. We have also carefully considered and addressed each comment in the revised manuscript.

**General:**

When the influence of assimilation frequency is investigated, it is not clear how data assimilation experiments with a longer assimilation time window of 3-hr are carried out. Are the hourly surface station observations averaged inside the 3-hr time window? Are the satellite data inside the 3-hr time window assumed to be valid at one particular instance? If so, when are they supposed to be valid?

Response: In Exp 5–8, we reduce the number of assimilated observations and only assimilate the daytime synthetic observations at discrete 3 h time steps (*i.e.*, 01:00, 04:00, 07:00, and 10:00 UTC) of the GEOS-Chem model rather than taking the average inside a 3 h time window. The satellite observations are assumed to be valid and assimilated at the corresponding hour when available.

The authors found that sometimes the data assimilation has negative effects. For instance, "In Japan and Mongolia, the assimilation of GEMS data generally contributes to a deterioration of simulated ozone and even counteracts the positive impact of surface observations when performing the joint assimilation." It is not impossible to encounter such cases. When this happens, it is probably worth to investigate the reason for such a behavior. With the current OSSE setting, it is probably not too hard to investigate the underlying causes.

Response: Accepted.

We have revised this part in the revised Section 3.1 (lines 299–303) as follows:

"However, the influence of assimilation efforts is complicated in East Asia, such as in Japan and Mongolia where the *a priori* ozone and its bias are relatively low. In this case, adding the synthetic GEMS observations results in a slight deterioration of simulated ozone and even counteracts the positive impact of surface observations when performing the joint assimilation. We attribute this partly to the improper specification of model errors and the spatial spread of observational information *via* transboundary transport."

**Specific:**

Line 20: It is probably better to replace "data assimilation better represents" to "data assimilation improves"

Response: Corrected.

We have replaced "better represents" with "improves". Please see line 20 in the revised manuscript.

Line 22: RMSE is a accuracy metric rather than a precision measure.

Response: Corrected.

We have replaced "precision improvements" with "significant improvements". Please see line 22 in the revised manuscript.

Line 113: Is "optimal estimation" the same as "optimal interpolation"?

Response: Optimal estimation is a regularized matrix inverse method based on Bayes' theorem. It is commonly used in geosciences to solve different kinds of inverse problems, particularly for atmospheric sounding (Rodgers, 2000). Optimal interpolation, however, is a relatively simple but useful method of data assimilation, which is a particular kind of inverse problem (Brasseur and Jacob, 2017).

Equation 5: It would be better to use "y" for variables in observation space.

Response: Corrected.

We have replaced "$\hat{x}^{obs}$" with "$y$" in the revised Section 2.3. Please see lines 196–197 in the revised manuscript.

Line 196: Xap should be in a vector in observation space, but it appears as a state vector. It is better to clearly differentiate state and observation vectors.

Response: Accepted.

$x_{ap}$ is the *a priori* profile (vector) used in the satellite retrieval procedure (Eq. 1 in Section 2.2). In the data assimilation procedure, the *a priori* profile $x_{ap}$, and averaging kernels $A$ from satellite retrievals (Section 2.2) are used for the calculation of observation operator $H$ for satellite measurements as follows. This procedure aims to remove the dependence of the analysis on the model-retrieval comparison (Miyazaki *et al.*, 2012, 2020) following our previous work (Shu *et al.*, 2022).

$$Hx^b = x_{ap} + A(Sx^b - x_{ap})$$

We have added the definition of the state vector ($x$) and use $y$ for the observation vector in the revised Section 2.3 (lines 195–196) to differentiate state and observation vectors as follows:

"At each assimilation time step, we calculate the optimal estimate $\hat{x}^a$ of the true ozone concentrations ($x$, state vector) as a weighted average of the model forecast $x^b$ and the observation $y$."

Line 211: It is reasonable to assume no correlation between surface station observations. But it is probably questionable to assume no correlation for satellite observations.

Response: Accepted.

Since the horizontal resolution of all the synthetic observations (satellite and surface observations) is much finer than that of the GEOS-Chem model (0.5°×0.625°), we adopt the super-observation approach to produce more representative data and reduce the horizontal observation error correlations (Miyazaki *et al.*, 2012; Barré *et al.*, 2015; Ma *et al.*, 2019). The approach is to average the observations (including errors and averaging kernels) across each 0.5° latitude × 0.625° longitude bin. In addition, it is also computationally cheaper to use super-observations in satellite data assimilation. We have previously discussed this in Section 2.3.

We have rephrased this part in the revised Section 2.3 (lines 213–218) to avoid ambiguity as follows:

"**R** is the observation error covariance matrix, including the contributions from the measurement error and the representativeness error. Since the horizontal resolution of all synthetic observations (GEMS, LEO satellite, and surface observations) is much finer than that of the model, we apply a super-observation approach to produce more representative data and reduce the horizontal observation error correlations (Miyazaki *et al.*, 2012; Ma *et al.*, 2019). A super-observation is generated by averaging all the observations (including errors and averaging kernels) within the same 0.5° latitude × 0.625° longitude GEOS-Chem model grid. Thus, **R** is assumed to be diagonal, that is, the observation errors are not correlated."

Figure 6d: What do the two different shades of color represent in the lower two panels?

Response: Corrected.

We have revised the caption of Fig. 6 (lines 729–736) to make it clear as follows:

"Figure 6. Comparison of the averaged diurnal cycle of (a) surface ozone, (b) mean

bias, and (c) RMSE, as well as (d) the histogram (in percentage) of the temporal correlation coefficient of hourly ozone at validation grids (Fig. S4), as simulated by the *control run* (*a priori*, in blue) and three *assimilation runs* (Exp 1–3 in Table 1, in green, yellow, and red, respectively) relative to the *nature run* ("True", black line in panel a) for June 2020. In panel (d), the green, yellow, and red bars respectively represent the frequency (%) of the temporal correlation coefficient of simulated ozone between the three *assimilation runs* (Exp 1–3) and the *nature run*, which is in comparison to that of the *control run* (in blue). The blue bars are the same in the three sub-panels to better illustrate the improvements in the temporal correlation of simulated ozone relative to the *control run*. The mean values of the temporal correlation coefficient (colored the same as lines in panel a) at all validation grids are inset."

**Reference:**

Barré, J., Gaubert, B., Arellano, A. F. J., Worden, H. M., Edwards, D. P., Deeter, M. N., Anderson, J. L., Raeder, K., Collins, N., Tilmes, S., Francis, G., Clerbaux, C., Emmons, L. K., Pfister, G. G., Coheur, P.-F., and Hurtmans, D.: Assessing the impacts of assimilating IASI and MOPITT CO retrievals using CESM-CAM-chem and DART, J. Geophys. Res. Atmos., 120, 10,501–10,529, https://doi.org/10.1002/2015JD023467, 2015.

Brasseur, G. P., and Jacob, D. J.: Modeling of Atmospheric Chemistry, Cambridge University Press, Cambridge, 2017.

Ma, C., Wang, T., Mizzi, A. P., Anderson, J. L., Zhuang, B., Xie, M., and Wu, R.: Multiconstituent Data Assimilation With WRF-Chem/DART: Potential for Adjusting Anthropogenic Emissions and Improving Air Quality Forecasts Over Eastern China, J. Geophys. Res. Atmos., 124, 7393–7412, https://doi.org/10.1029/2019JD030421, 2019.

Miyazaki, K., Bowman, K. W., Yumimoto, K., Walker, T., and Sudo, K.: Evaluation of a multi-model, multi-constituent assimilation framework for tropospheric chemical reanalysis, Atmos. Chem. Phys., 20, 931–967, https://doi.org/10.5194/acp-20-931-2020, 2020.

Miyazaki, K., Eskes, H. J., Sudo, K., Takigawa, M., van Weele, M., and Boersma, K. F.: Simultaneous assimilation of satellite NO2, O3, CO, and HNO3 data for the analysis of tropospheric chemical composition and emissions, Atmos. Chem. Phys., 12, 9545–9579, https://doi.org/10.5194/acp-12-9545-2012, 2012.

Rodgers, C.: Inverse Methods for Atmospheric Sounding: Theory and Practice, 2000.

Shu, L., Zhu, L., Bak, J., Zoogman, P., Han, H., Long, X., Bai, B., Liu, S., Wang, D., Sun, W., Pu, D., Chen, Y., Li, X., Sun, S., Li, J., Zuo, X., Yang, X., and Fu, T.-M.: Improved ozone simulation in East Asia via assimilating observations from the first geostationary air-quality monitoring satellite: Insights from an Observing System Simulation Experiment, Atmos. Environ., 274, 119003, https://doi.org/10.1016/j.atmosenv.2022.119003, 2022.